# Oropouche Virus Infects, Persists and Induces IFN Response in Human Peripheral Blood Mononuclear Cells as Identified by RNA PrimeFlow™ and qRT-PCR Assays

**DOI:** 10.3390/v12070785

**Published:** 2020-07-21

**Authors:** Mariene Ribeiro Amorim, Marjorie Cornejo Pontelli, Gabriela Fabiano de Souza, Stéfanie Primon Muraro, Daniel A. Toledo-Teixeira, Julia Forato, Karina Bispo-dos-Santos, Natália S. Barbosa, Matheus Cavalheiro Martini, Pierina Lorencini Parise, Aline Vieira, Guilherme Paier Milanez, Luis Lamberti Pinto daSilva, Pritesh Jaychand Lalwani, Alessandro Santos Farias, Marco Aurélio Ramirez Vinolo, Renata Sesti-Costa, Eurico Arruda, Jose Luiz Proenca-Modena

**Affiliations:** 1Department of Genetics, Evolution, Microbiology and Immunology, Institute of Biology, University of Campinas, Campinas 13083-862, Brazil; mariene.ramorim@gmail.com (M.R.A.); gabriela.sfabiano@gmail.com (G.F.d.S.); stefaniemuraro@gmail.com (S.P.M.); teixeiradatt@gmail.com (D.A.T.-T.); foratojulia@gmail.com (J.F.); karina.bsantos@hotmail.com (K.B.-d.-S.); crmv2007@hotmail.com (M.C.M.); pierinalp@gmail.com (P.L.P.); allica.vieira@gmail.com (A.V.); guilhermemilanez@gmail.com (G.P.M.); farias.as@gmail.com (A.S.F.); mvinolo@unicamp.br (M.A.R.V.); 2Department of Cell Biology and Virology Research Center, Ribeirão Preto Medical School, University of São Paulo, Ribeirão Preto 14049-900, Brazil; marpontelli@gmail.com (M.C.P.); sbarbosanatalia@gmail.com (N.S.B.); lldasilva@fmrp.usp.br (L.L.P.d.); 3Leônidas and Maria Deane Institute (ILMD), Fiocruz Amazônia, Manaus 69057-070, Brazil; lalwanipritesh@yahoo.com; 4Hematology and Hemotherapy Center, Medical School of the University of Campinas, Campinas 13083-887, Brazil; renata.sesti@gmail.com

**Keywords:** Oropouche virus, PBMC, lymphocyte, interferons, RNA PrimeFlow™, dendritic cells, monocytes, B cells

## Abstract

*Oropouche orthobunyavirus* (OROV) is an emerging arbovirus with a high potential of dissemination in America. Little is known about the role of peripheral blood mononuclear cells (PBMC) response during OROV infection in humans. Thus, to evaluate human leukocytes susceptibility, permissiveness and immune response during OROV infection, we applied RNA hybridization, qRT-PCR and cell-based assays to quantify viral antigens, genome, antigenome and gene expression in different cells. First, we observed OROV replication in human leukocytes lineages as THP-1 monocytes, Jeko-1 B cells and Jurkat T cells. Interestingly, cell viability and viral particle detection are maintained in these cells, even after successive passages. PBMCs from healthy donors were susceptible but the infection was not productive, since neither antigenome nor infectious particle was found in the supernatant of infected PBMCs. In fact, only viral antigens and small quantities of OROV genome were detected at 24 hpi in lymphocytes, monocytes and CD11c^+^ cells. Finally, activation of the Interferon (IFN) response was essential to restrict OROV replication in human PBMCs. Increased expression of type I/III IFNs, ISGs and inflammatory cytokines was detected in the first 24 hpi and viral replication was re-established after blocking IFNAR or treating cells with glucocorticoid. Thus, in short, our results show OROV is able to infect and remain in low titers in human T cells, monocytes, DCs and B cells as a consequence of an effective IFN response after infection, indicating the possibility of leukocytes serving as a trojan horse in specific microenvironments during immunosuppression.

## 1. Introduction

Arboviruses, or arthropod-borne viruses, such as *Dengue virus* (DENV), *Japanese encephalitis virus* (JEV), *West Nile virus* (WNV), *Chikungunya virus* (CHIKV) and *Zika virus* (ZIKV), are major agents of global public health concern. These agents are responsible for causing diseases with high morbidity and mortality rates in developed and nondeveloped countries. *Oropouche orthobunyavirus* (OROV) and *Mayaro virus* (MAYV) are two emerging viruses that have been reported as candidates to the next big epidemics in countries from South America, such as Brazil [1,2].

OROV is an emerging virus that causes a Dengue-like illness known as Oropouche fever, occurring especially in the Amazon region of Brazil, Peru and Venezuela. More than 500,000 cases have been reported since its discovery in 1955 including recent cases out of the Amazon region in Brazil [3,4,5,6]. The major symptoms include headache, myalgia, arthralgia, malaise, photophobia, exanthema and polyuria. An interesting fact is that symptoms can reappear one or two weeks after recovery in about 60% of the patients. Additionally, there are reports of patients that showed hemorrhagic symptoms or neurological complications associated with OROV detection [7,8,9]. OROV has a high potential of dissemination that is associated with environmental and anthropological factors, such as high population densities, deforestation and changes in natural vegetation. *Culicoides paraensis* is reported as the main vector in the urban cycle of viral transmission [2,8,10]. Regarding the virus taxonomy, OROV is a member of the *Peribunyaviridae* family, *Orthobunyavirus* genus and is part of Simbu serogroup. Similar to other orthobunyaviruses, it is an enveloped virus with tri segmented single-stranded negative-sense RNA genome. The large segment (L) encodes the viral RNA-dependent RNA polymerase. The medium segment (M) encodes a polyprotein that later originates the envelope glycoproteins Gc and Gn, and the nonstructural protein NSm. The small segment (S) encodes the nucleocapsid protein (N) that protects viral RNA from degradation and other nonstructural protein NSs. OROV NSm and NSs proteins have started to be explored by the generation of OROV recombinant viruses, and NSs favored viral replication in A549 cells [11,12].

The innate immune response to viral infections plays an important role to contain viral replication. At the early moments of a viral infection, cellular intrinsic components known as pattern recognition receptors (PRRs) recognize pathogen-associated molecular patterns (PAMPs) triggering signaling cascades that lead to the production of a variety of cytokines, especially interferons (IFNs), which are essential to contain viral replication. OROV negative genome is an important PAMP that can be detected by Toll-like receptors (TLR) at the endosome membrane or RIG-like receptors in the cytoplasm [11,13]. Antigen-presenting cells (APCs) circulating in the blood have a high expression of these PRRs and consequently play an important role in recognizing PAMPs and triggering IFN response by interferon-stimulated genes (ISGs). APCs are also important to the development of adaptive immunity [14,15].

Some viruses have mechanisms that allow them to evade this early response, leading to disease. It was recently reported the capacity of ZIKV to infect human dendritic cells (DCs), modulating the expression of important antiviral genes, such as type I IFN, genes and TLRs genes, and maturation markers as *CD80* and *CD86* genes [16]. ZIKV is also capable of infecting human circulating monocytes and affecting their inflammatory response [17]. On the other hand, the immune response against OROV infection has started to be explored recently. Studies in animal models have shown an important role for TLR3, MAVS and type I IFN against OROV infection and neuropathogenesis. Moreover, in vitro infection of DCs and phagocytes is facilitated in animals lacking these genes [18]. Additionally, OROV proteins were detected in cells from peripheral blood of patients in the acute phase of Oropouche fever [3].

Little is known about the OROV capacity of infecting human peripheral blood mononuclear cells (PBMCs) and their response to this event. In the present study, we evaluated OROV capacity to infect, persist and the pattern of an innate immune response of human lineages of leukocytes and human PBMCs in different moments after infection. Our results show that OROV infect, replicate and is able to persist in human Jurkat T cells, THP-1 monocytic cells and Jeko-1 Mantle lymphoma B cells. However, OROV has a low capacity for replicating in human PBMCs from healthy donors. Focus forming assays were able to detect low quantities of OROV in PBMCs supernatants after 48 h of infection, in addition to an increase in the expression of type I, II and III IFNs and ISGs. Even though components of the innate immune response apparently contain the infection in vitro, it was observed using RNA PrimeFlow™ and flow cytometry that viral genome was still found inside a few populations of human monocytes and lymphocytes, indicating the possibility of these cells serving as a trojan horse in specific microenvironments, mainly in immunosuppressive conditions, since viral replication is re-established when type I IFN receptor (IFNAR) is blocked or the cells are treated with glucocorticoid.

## 2. Materials and Methods

### 2.1. Ethics and Samples Collection

Ethical approval was obtained from the Ethics and Research Committee of the University of Campinas (Certificate of Presentation for Ethical Consideration [CAAE] No. 89082718.5.0000.5404), in August 26, 2018. Informed consent was signed by the individuals that agreed to participate as healthy donors.

After signing the informed consent, 9 mL of peripheral whole blood were collected in heparinized tubes by venipuncture, in the Laboratory of Emerging Viruses at the University of Campinas (Unicamp), Brazil. This study included people from 18 to 30 years old, without febrile disease reported in the last few months, neither under anti-inflammatory nor antimicrobial medications.

### 2.2. Reagents

Leibovitz medium (Sigma-Aldrich, Darmstadt, Germany); penicillin–streptomycin solution (Sigma-Aldrich, Darmstadt, Germany); MEM (Earle’s Minimum Essential Medium) (Sigma-Aldrich, Darmstadt, Germany); Fetal Bovine Serum (FBS) (Gibco, Thermo Fisher Scientific, Waltham, MA, USA); N-(2-Hydroxyethyl)piperazine-*N*′-(2-ethanesulfonic acid; HEPES) solution (Sigma-Aldrich, Darmstadt, Germany); RPMI 1640 (Sigma-Aldrich, Darmstadt, Germany); Trypan-blue 0.4% (Sigma-Aldrich, Darmstadt, Germany); Ficoll-Paque (GE Healthcare Biosciences, Pittsburgh, PA, USA) 1.077 g/mL; Dexamethasone (Sigma-Aldrich, Darmstadt, Germany); Anti-Interferon alpha/beta receptor 1 antibody (Abcam, Cambridge, UK); Trizol^®^ plus (Invitrogen, Thermo Fisher Scientific, Waltham, MA, USA) reagent; carboxymethylcellulose (Sigma-Aldrich, Darmstadt, Germany); Triton^®^ X-100 (Merck KGaA, Darmstadt, Germany); Oropouche immune ascitic fluid ATCC^®^ VR-1228AF™ (American Type Culture Collection [ATCC], Manassas, VA, USA ); Antimouse IgG (whole molecule)–Peroxidase antibody produced in goat (Sigma-Aldrich, Darmstadt, Germany); TrueBlue™ Peroxidase Substrate (KPL, SeraCare, Gaithersburg, MD, USA ); PrimeFlow™ RNA Assay Kit (Invitrogen, Thermo Fisher Scientific, Waltham, MA, USA); Fixable Viability Stain 450 (BD Biosciences, San Jose, CA, USA); anti-CD14 PE, anti-HLA-DR APC-H7, HLA-DR PerCP, anti-CD3 V500, anti-CD19 PE CF594, anti-CD11c APC, CD8 APC-Cy7 (BD Biosciences, San Jose, CA, USA); Mounting Medium With DAPI - Aqueous, Fluoroshield (Abcam, Cambridge, UK); Cytofix/Cytoperm™ (BD Biosciences, San Jose, CA, USA), FITC Annexin V and Propidium Iodide (PI) (BioLegend Inc., San Diego, CA, USA); Anti-mouse IgG conjugated with Alexa Fluor 594, Anti-mouse IgG conjugated with Alexa Fluor 488, Anti-rabbit IgG conjugated with Alexa Fluor 647 (Abcam, Cambridge, UK); Rabbit Anti-TGN46 polyclonal antibody (Merck Millipore, Burlington, MA, USA).

### 2.3. Virus and Cells

The present study was developed using the OROV BeAn19991 strain, provided by Prof. Dr. Luiz Tadeu Morais Figueiredo from the Medical School of the University of São Paulo, Ribeirão Preto, Brazil. Vero ATCC CCL81 and C6/36 ATCC CRL 1660 cells (ATCC, Manassas, VA, USA) were used to propagate the virus. C6/36 cells were propagated in Leibovitz medium supplemented with 1% penicillin–streptomycin solution (10,000 UI/mL penicillin and 200 mM streptomycin), 10% FBS and incubated at 28 °C to reach 70% confluence. Vero CCL81 cells were propagated in MEM supplemented with 10% FBS, 1% PS and incubated at 37 °C, 5% CO_2_ atmosphere. To produce virus stock, virus was added to a 75 cm^2^ tissue culture bottle (Nest Biotechnology Co., LTD, Wuxi New District, China) with C6/36 cells monolayer. Leibovitz medium supplemented with 2% FBS and 1% PS was added 2 h post-infection (hpi). The cells were incubated at 28 °C for 5 days. Cells and supernatants were collected and quickly frozen in liquid nitrogen and stored at −80 °C. This stock was previously clarified by centrifugation (10,000× *g* for 5 min) for use. A second passage of OROV was made by propagating the previously produced stock in Vero CCL81 cells with MEM supplemented with 1% PS without FBS, incubated at 37 °C, 5% CO_2_ atmosphere for 3 days, when the cells showed 70% of cytopathic effect (CPE). Cells and supernatants were collected and quickly frozen in liquid nitrogen and stored at −80 °C. This stock was also clarified by centrifugation before use. The titers of the stocks were determined by focus forming assay (FFA) per mL [18].

THP-1 ATCC TIB-202 and Jurkat ATCC TIB-152 cells were kindly donated by Prof. Rafael Elias Marques Pereira Silva from Brazilian Biosciences National Laboratory (LNBio), National Center for Research in Energy and Materials (CNPEM), Campinas, São Paulo, Brazil. These cells were propagated in RPMI 1640 medium supplemented with 10% FBS and 1% PS. Jeko-1 Mantle cell lymphoma CRL-3006-ATCC (ATCC, Manassas, VA, USA) were propagated in RPMI medium supplemented with 20% FBS and 1% PS. The suspension of cells was incubated at 37 °C, 5% CO_2_ atmosphere. For the tests with OROV, lymphocytes were counted with Trypan-blue 0.4% and seeded in 12- and 24-well plates (Nest Biotechnology Co., LTD, Wuxi New District, China). Infection was proceeded with a first step of centrifugation (400× *g* for 10 min) [19], following incubation at 37 °C, 5% CO_2_ atmosphere for 2 h. Then, cells were washed, and RPMI 1649 medium 2% SFB and 1% PS were added. Cells were incubated at 37 °C, 5% CO_2_ atmosphere for infection kinetics assays.

### 2.4. Human PBMC Culture and OROV Infection

PBMC were obtained by gradient centrifugation with Ficoll-Paque 1.077 g/mL, according to the manufacturer’s instructions with some modifications. Briefly, whole blood was diluted 1:1 in RPMI 1640 serum-free medium. This dilution was carefully placed over the Ficoll-Paque solution 1:3, in a centrifuge tube (Nest Biotechnology Co., LTD, Wuxi New District, China), which was then centrifuged 300× *g* for 35 min at room temperature (RT) with slow acceleration and break off. The cells were washed twice with phosphate-buffered saline 0.15 M, pH7.4 (PBS 1×), counted with Trypan-blue 0.4% and seeded into 24-well (4 × 10^5^ cells per well) ultralow attachment surface culture plates (CORNING Inc., Lowell, MA, USA) with RPMI 1640 medium supplemented with 1% PS. For OROV infection, virus stock was added to a final volume of 200 to 300 µL, and Multiplicity of Infection (MOI) 0.1, 1 and 10. The plates were carefully placed at the centrifuge for a first step of centrifugation (300× *g* for 10 min) at 32 °C, following incubation at 37 °C, 5% CO_2_ atmosphere for 2 h. After that, the wells were washed with PBS 1× and new media supplemented with 2% FBS and 1% PS was added. Cells were incubated at 37 °C, 5% CO_2_ atmosphere. in some experiments human PBMCs were seeded in 24-well ultralow attachment surface and pretreatment with Dexamethasone (Dx) 1 µM or Anti-Interferon alpha/beta receptor 1 (Anti-IFNAR1) antibody 5 ng/mL for 2 h [20,21]. Then, virus stock was added to a final volume of 200 to 300 µL, and MOI 1. The plates were incubated and 2 hpi the medium with virus was washed out, and new media supplemented with 2% FBS and 1% PS, Dx 1 µM or anti-IFNAR1 5 ng/mL was added. Cells were incubated at 37 °C, 5% CO_2_ atmosphere. All controls received supernatants of noninfected Vero cells.

### 2.5. Whole Blood Infection

Whole blood obtained from healthy donors in heparinized tubes were diluted 1:1 in RPMI 1640 medium supplemented with 1% PS and 1% HEPES, and infected with OROV as previously described [17]. The control received supernatants of the Vero cell culture instead of the virus. Aliquots were taken at times 0, 2, 4, 12, 24 and 48 hpi for RNA extraction with QIAamp RNA blood Mini Kit (QIAGEN, Thermo Fisher Scientific, Waltham, MA, USA) following the manufacturer’s instructions. Cell viability was assessed by flow cytometry using FITC Annexin V and Propidium Iodide (PI) following the manufacturer instructions.

### 2.6. RNA Extraction and qRT-PCR for OROV Quantitation

Cells were centrifuged (400× *g* for 10 min), and the pellets were resuspended in 300 µL of Trizol^®^ Plus reagent and stored at −80 °C. The supernatants were quickly frozen in liquid nitrogen and stored at −80 °C. PureLink™ RNA Mini Kit (Invitrogen, Thermo Fisher Scientific, Waltham, MA, USA) was used for RNA extraction of cell lysates, according to the manufacturer’s instructions. Additionally, PureLink™ Viral RNA/DNA Mini Kit (Invitrogen, Thermo Fisher Scientific, Waltham, MA, USA) was used for RNA extraction from the supernatants, according to the manufacturer’s instructions. RNA samples were stored at −80 °C. The qRT-PCR was performed using TaqMan^®^ Fast Virus 1-Step Master Mix (Applied Biosystems, Thermo Fisher Scientific, Waltham, MA, USA) following the manufacturer’s instructions. Cycles consisted of one step of reverse transcription at 50 °C for 5 min and 95 °C for 20 s, 45 cycles of amplification steps at 95 °C for 5 s and annealing–extension at 60 °C for 30 s. All determinations were performed in triplicate. We used a QuantStudio™ 3 Real-Time PCR System, 96-well, 0.1 mL (Applied Biosystems, Thermo Fisher Scientific, Waltham, MA, USA) thermal cycler. Primers and probe sequences used for OROV detection are described in Table 1 [18]. For analysis, a standard curve was made using a previously titrated OROV, and the results are represented in Log10 FFU equivalents/mL.

### 2.7. qRT-PCR for Gene Expression

Total RNA was reverse transcribed using the High Capacity cDNA Reverse Transcription kit (Applied Biosystems, Thermo Fisher Scientific, Waltham, MA, USA) according to the manufacturer’s instructions. After cDNA quantification by NanoDrop One (Thermo Fisher Scientific, Waltham, MA, USA), qRT-PCR was performed using Universal iTaq SYBR Green Supermix (Bio-Rad Laboratories, Hercules, CA, USA) following the manufacturer’s instructions and using a QuantStudio™ 3 Real-Time PCR System. We used 5 µL SYBR Green Mix (2×), 3 µL of 600 nM forward and reverse primers mix (Thermo Fisher Scientific, Waltham, MA, USA) and 2 µL of 90–100 ng/mL cDNA sample per reaction. Cycles consisted of polymerase activation and DNA denaturation at 95 °C for 3 min, 40 cycles of amplification and annealing–extension at 95 °C for 15 s followed by 60 °C for 1 min, respectively. All primers used for gene expression analysis are described in Table 1. GAPDH was used as a housekeeping gene to calculate the relative quantification by the ∆∆*C*_T_ method.

### 2.8. Focus Forming Assay (FFA)

Supernatants from infected cultures were analyzed by FFA, performed as previously described [18,33] with some modifications. Semi-solid medium (MEM with final concentrations of carboxymethylcellulose 0.75%, FBS 2% and PS 1%) was added 2 h post-infection. The fixation process was performed 72 hpi with paraformaldehyde 1% overnight at 4 °C. Cell monolayers were washed 3× with PBS 1×, and blocked with 5% milk in PBS 1x (blocking solution) at room temperature for 30 min. Cells were permeabilized with a perm wash buffer (PBS 1×, BSA 0.1%, Triton^®^ X-100 0.1%) by washing 3×. Then, Oropouche immune ascitic fluid was used (1:1000 in blocking solution) as the primary antibody. Antimouse IgG (whole molecule)–Peroxidase antibody was added (1:2000 in blocking solution). Foci were developed with TrueBlue Peroxidase Substrate and counted visually. All determinations were performed in triplicate.

### 2.9. Single-Cell OROV Detection by RNA PrimeFlow™

RNA probes to detect OROV were designed to align at segment L genome and antigenome. First, Vero CCL81 cells were infected with OROV with both MOI 1 and MOI 10. RNA detection by RNA PrimeFlow™ assay was performed as indicated by the manufacturer. Briefly, infected cells were washed twice with FBS 2% in PBS 1×, fixed and permeabilized with PrimeFlow reagents. Then, RNA probes for genome (VF1-6000635) and antigenome (VF4-6000634) were added for hybridization at 40 °C. After that, three steps of signal amplification were performed: preamplification, amplification and label probes hybridization. Infection kinetics in Vero cells were also evaluated by qRT-PCR and FFA. Infected THP-1, Jurkat and Jeko-1 cells were also submitted to RNA PrimeFlow™ protocol described above. And finally, human PBMC infected in vitro were submitted to RNA PrimeFlow™ protocol, with some modifications. First, cells were stained for viability using BD Horizon™ Fixable Viability Stain 450, according to the manufacturer’s protocol. After that, cells were washed with FBS 2% in PBS 1× and incubated at 4 °C with antibodies for surface markers anti-CD14 PE, anti-HLA-DR APC-H7, anti-CD3 V500 and anti-CD19 PE CF594. Finally, cells were submitted to the RNA PrimeFlow™ protocol. Samples were analyzed in a BD FACSVerse flow cytometer (BD Biosciences, San Jose, CA, USA).

### 2.10. OROV Proteins, Genome and Antigenome Detection by Confocal Microscopy

We used RNA PrimeFlow™ assay reagents to standardize a protocol for intracellular detection of OROV genome/antigenome by immunofluorescence. Basically, we followed the manufacturer’s instructions for flow cytometry, with some modifications. Vero cells were seeded on 13 mm diameter coverslips (Knittel-Glass, Braunschweig, Germany) in 24-well plates (2 × 10^4^ cells/mL) and infected with OROV. Nonadherent cells such as leukocytes lineages and human PBMC were seeded in 8 wells Lab-Tek^®^ Glass Chamber Slide™ (Thermo Fisher Scientific, Waltham, MA, USA) in PBS 1x and incubated at 37 °C for 30 min [34]. Then, PBS was removed for the addition of media with virus inoculum. After the indicated time post-infection, the coverslips and the wells were washed with PBS 1x and fixed with PrimeFlow^®^ RNA Fixation buffer 1 for 1 h at 4 °C. Then, cells were permeabilized with PrimeFlow^®^ RNA Permeabilization Buffer and incubated with Oropouche immune ascitic fluid (1:50 in permeabilization buffer) for 1 h at 4 °C. This step was also performed with anti-TGN46 primary antibody (1:50 in permeabilization buffer) in some experiments. The coverslips were washed with PrimeFlow^®^ RNA Wash Buffer once and PrimeFlow^®^ RNA Permeabilization Buffer twice. Then we added secondary antibodies (1:1000 in permeabilization buffer) and incubated for 30 min at 4 °C. We performed a second step of fixation with PrimeFlow^®^ RNA Fixation Buffer 2 for 1 h. Coverslips were washed with PrimeFlow^®^ RNA Wash Buffer containing RNase Inhibitors and incubated with target probes (1:20 in PrimeFlow^®^ RNA Target Probe Diluent) for 2 h at 40 °C. After amplification, we washed the coverslips and the wells with MilliQ water 3 times. Finally, we prepared the slips for confocal microscopy with Fluoroshield Mounting Medium with DAPI following the manufacturer’s instructions. Confocal microscopy was performed at the Central Laboratory of High-Performance Technologies (LaCTAD) at the University of Campinas, Campinas, Brazil, using a Leica TCS SP5 II confocal microscope (Leica Microsystems, Wetzlar, Germany).

### 2.11. Immunofluorescence and Flow Cytometry

PBMC and polymorphonuclear cells (PMN) were infected with OROV at MOI 1 in suspension and 24 hpi, cells were first stained for immunophenotyping using human anti-CD14 and CD11c. Then cells were fixed and permeabilized using Cytofix/Cytoperm solutions according to the manufacturer. For OROV staining, we used Oropouche immune ascitic fluid polyclonal antibodies and anti-mouse 488. Cells that were used to immunofluorescence were put on glass coverslips previously treated with l-polylysine. Immunofluorescence images were obtained using a confocal microscopy Leica TCS SP8. For flow cytometry phenotyping, ten thousand cells were analyzed by FACS Calibur (BD Biosciences, San Jose, CA, USA) using the CellQuest program (BD Biosciences, San Jose, CA, USA).

### 2.12. Graphics and Statistical Analysis

All the molecular assays and FFA assays were performed in triplicates. Media, Standard Error (SEM) and the statistical analysis were conducted using the GraphPad Prism 5 software (Graphpad Software, San Diego, CA, USA). Flow cytometry data were analyzed by FlowJo V10 (FlowJo LLC, BD Biosciences, San Jose, CA, USA). For viral load analysis, the log titers were analyzed by the Mann-Whitney test or two-way ANOVA. The relative expression of different genes determined by qRT-PCR were also compared using two-way ANOVA and Dunnett’s multiple comparisons test, or one-way ANOVA and Dunn’s multiple comparisons test. *p* < 0.05 was accepted as statistically significant in all experiments.

## 3. Results

### 3.1. Detection of OROV Genome and Antigenome by RNA Primeflow™ Assay

In order to evaluate which cells are productively or persistently infected with OROV, we decided to standardize the RNA PrimeFlow™ assay to identify the OROV genome and antigenome at a single-cell level, since antigenome detection is an unambiguous biomarker of viral replication. In addition, this technique allows a simultaneous identification of RNA and proteins occurring in a single event by flow cytometry. Thus, to validate this assay for OROV and to verify OROV RNA probes functionality, we compared RNA PrimeFlow results with other techniques after OROV infection in Vero cells. The workflow of the methodology is summarized in Appendix A. Previous studies have shown that OROV is able to infect and replicate in these cells [2,18]. As expected, it was possible to detect the OROV genome and antigenome increasing over the time of infection by flow cytometry after the RNA PrimeFlow technique (Figure 1A), indicating productive viral replication. These results were corroborated by focus forming assay (FFA) and qRT-PCR results obtained from the same samples of cells (Figure 1B,C). Finally, to validate our genome and antigenome staining, we used an immunofluorescence assay based on the RNA PrimeFlow™ protocol. As predicted, the RNA PrimeFlow staining was specific since it was possible to observe viral antigenome colocalizing with OROV proteins in perinuclear regions of infected cells (Figure 1D).

In addition, OROV genetic material and viral antigens colocalizing with trans-Golgi network marker TNG46 (Appendix A), which is in agreement with previously published studies showing that the Golgi complex is the main assembly site for OROV [35]. Thus, this innovative technique based on RNA hybridization was considered adequate to differentiate cells in which OROV is only internalized from those in which OROV is able to replicate.

### 3.2. OROV Productively Infects Different Human Leukocytes Lineages

Previous studies demonstrated a productive infection of mononuclear cells by OROV in an immunodeficient mouse model [18]. Additionally, OROV antigens were detected in peripheral blood leukocytes from patients with Oropouche fever by indirect immunofluorescence [3]. Thus, we aimed to investigate OROV capacity to infect different lineages of human leukocytes. We infected Jurkat, Jeko-1 and THP-1 cells with OROV at MOI 1, and collected the supernatants and cell lysates for analysis in different moments after infection. Results from qRT-PCR showed the maintenance of the viral genome in cell lysates and supernatant through 48 hpi in all cell lines tested (Figure 2A). Moreover, OROV productive infection was only observed in Jurkat T cells over the time of infection, while in monocytic THP-1 cells and Jeko-1 B cells, the production of infective particles decreased over time (Figure 2B). Next, to evaluate the intracellular pattern of viral infection, Jurkat and Jeko cells were infected with OROV and submitted to the hybridization assay followed by confocal microscopy imaging. Jurkat and THP-1 were also infected and submitted to the hybridization assay followed by flow cytometry As expected, the frequency and signal intensity were higher in Jurkat cells, in which it was possible to identify viral genome clusters colocalizing with viral proteins around the cytoplasm, especially on the borders of the cell, while the antigenome was closer to the nucleus, at 48 hpi. Only a few Jeko-1 infected cells were observed by confocal microscopy (Figure 2C). Consistently, we found a higher percentage of Jurkat OROV gRNA and OROV agRNA positive cells than THP-1 by flow cytometry (Figure 2D). Regardless of how productive the infection was, all cell lineages analyzed in this study were able to maintain high intracellular levels of viral RNA, with viral particles detected in the supernatant even after subculturing these cells three times (Figure 2E). These data highlight the possibility of OROV infecting human blood mononuclear leukocytes, with the maintenance of the genome in the cells even after cellular division. In addition, OROV infection is apparently more productive in human T cells than in B cells or monocytes.

### 3.3. Human Peripheral Blood Leukocytes Are Susceptible to OROV Infection but Do Not Generate Large Quantities of Infectious Particles

Some studies have demonstrated that other arboviruses, such as ZIKV and YFV, are capable of infecting human peripheral blood cells [17,36]. Thus, we set out to investigate OROV infection in leukocytes populations obtained from healthy donors. We first analyzed the OROV in vitro infection in whole blood, a previously described model that simulates the human in vivo infection [17]. Although some genes related to innate immune response were significantly upregulated after OROV infection (Appendix A), viral RNA production, rate of infected positive cells and type I IFN production in whole blood was very low. Moreover, since we observed apoptotic cells early in the culture (Appendix A), we decided to move to other models of infection. Next, we performed experiments using PBMCs isolated from healthy donors, which were infected with OROV in different MOI in vitro. In order to characterize which cell types were productively infected with OROV, cells and supernatants were collected at 1, 6, 12, 24 and 48 hpi to be analyzed by qRT-PCR, FFA, RNA PrimeFlow and conventional FACs assays. Interestingly, viral RNA was observed through 48 hpi with MOI 1 and 10 in cell lysates while viral.

RNA levels decreased over time in PBMCs supernatant (Figure 3A). Consistently, the results obtained by FFA revealed that only a few infectious particles are released in the supernatant until 12 hpi (Figure 3B). Finally, the RNA PrimeFlow™ flow cytometry data showed an increasing frequency of CD3^+^ and CD14^+^ cells carrying the viral genome (Figure 3C), indicating that human T lymphocytes and monocytes can be infected by OROV, even in environments where different human leukocytes are present. Next, in order to evaluate if the intracellular morphology of the viral factories were similar to what is observed in cells knowingly susceptible and permissive to OROV, infected and noninfected PBMCs were submitted to the hybridization assay followed by confocal microscopy imaging. Consistent with what has been observed in Vero cells, leukocytes infected cells from PBMCs showed viral gRNA, agRNA and antigens in the perinuclear regions, where the viral factories of OROV are usually mounted (Figure 3E).

In agreement with our previous results, OROV antigens were detected in monocytes (MC, CD14^+^), dendritic (DC, CD11c^+^ CD11b^+^) and polymorphonuclear (PMN, CD11c^+^CD11b^neg^) cells after infection of peripheral blood cells from healthy donors (Appendix A). In fact, despite the individual differences between the number of total cells, the PMNs, DCs and MCs from all three donors were susceptible to OROV (Appendix A) and intense labeling throughout the cytoplasm of these cells was observed after immunofluorescence staining (Appendix A). Thus, together, these results suggest that the myeloid cells are susceptible and permissive to OROV infection.

### 3.4. Human PBMCs Response to OROV Infection Involves Upregulation of IFNs and ISGs Expression

The recognition of viral PAMPs such as single-stranded RNA (ssRNA) and double-stranded RNA (dsRNA) in the endosomes and cytoplasm respectively by PRRs triggers signal cascades that result in the activation and migration of transcription factors to the nucleus, which induces the expression of cytokines and IFNs (Figure 4A). We hypothesized that viral genome maintenance without a large production of infectious particles could be a consequence of an effective innate immune response in PBMCs. Evaluation of IFNs transcription by qRT-PCR over the time of infection showed a crescent increase in type I IFN α and β mRNA expression that reached 50-fold change at 48 hpi in comparison to time 0 (noninfected cells). Interestingly, type III IFN also significantly increased, in 100-fold at 48 hpi (*p* < 0.0001), while type II IFN increased to almost 50-fold at 24 hpi (*p* < 0.0001) and slightly decreased 48 hpi (Figure 4B). Corroborating with these data, we found a crescent increase of the ISGs Mx1 expression at 12 hpi (*p* < 0.0001) and IFIT1 expression at 12 and 48 hpi (*p* < 0.0001; Figure 4D). Moreover, the mRNA expression of RLR such as MDA5, which is a RIG-I-like receptor dsRNA helicase enzyme, significantly increased at 12 and 24 hpi (*p* < 0.0001) and the TLR3 adapter molecule TRIF also increased, especially at 48 hpi (*p* < 0.0001; Figure 4C). Finally, we also observed a rapid induction of IL-6 and TNFα expression that could be due to APCs, T and B cell responses (Figure 4E) [37]. Altogether, these data provide evidence of the known immune response to RNA viruses, here a negative-sense RNA virus of the Peribunyaviridae family. This response was more prominently observed after 24 hpi with the induction of proinflammatory cytokines as soon as 6 hpi, and IL-10 anti-inflammatory cytokine as soon as 12 hpi.

### 3.5. IFNAR Blocking and Glucocorticoid Treatment Favor OROV Replication in PBMCs

The IFN response to viral infections is an important mechanism for containing initial virus spread. The importance of the type I IFN signaling pathway has been demonstrated for different viruses, including ZIKV, in mice models [38,39]. The host immune response to OROV infection also relies on the IFN signaling pathway [18,40]. To evaluate the importance of type I IFN in controlling OROV infection in human PBMCs, we pretreated cells with the anti-IFNAR antibody for 2 h, and infected them with OROV at MOI 1 for 48 h. We observed that, until 48 hpi, viral RNA is maintained in treated and nontreated cells (Figure 5A). However, higher quantities of viral RNA was detected in the supernatant treated in contrast to nontreated cells (Figure 5B). Additionally, FFA results showed infectious particles being released in the supernatant and reaching 10^4^ FFU/mL (Figure 5C). It is known that PBMCs treatment with Dexamethasone in vitro decreases the levels of cytokines in the culture [21]. Then, we did a 2 h pretreatment of the PBMCs with Dexamethasone and infected the cells with OROV at MOI 1 to evaluate viral replication. The treatment also favored viral replication and production of infectious particles, however in a lower magnitude than the anti-IFNAR treatment (Figure 5A–C). These results demonstrate the importance of the early events of host innate immune response to OROV infection in the peripheral blood, including type I IFN and cytokines release after pathogen recognition. In addition, the capacity of OROV to persist in different types of human leukocytes in association with an effective innate immune response and their ability to be reactivated after blocking the IFN pathway, indicate the possibility of leukocytes serving as a trojan horse in specific microenvironments during events of immunosuppression.

## 4. Discussion

Oropouche fever is an emergent disease with an increasing impact in Brazil and other countries in South America. The disease that has affected more than half a million people can also lead to neurological complications. OROV publications have increased in the last few years, as the number of arbovirus reports has also increased [8]. However, little is known about the human immune response to OROV infection and its pathogenesis has just started to be elucidated [18]. In this study, we evaluated the OROV capacity of infecting human PBMCs in vitro, viral replication and innate immune response mediated by these cells. For that, we efficiently standardized an RNA PrimeFlow™ protocol with RNA probes to identify the OROV genome and antigenome in infected cells. We could also evaluate OROV formation of viral factories, where the RNA probes colocalized with the TNG46 protein, corroborating with a previous study [35], demonstrating the efficiency of the assay. This is an important approach to consider since the options of anti-OROV antibodies in the market are very restricted.

In the present work, we applied RNA hybridization to characterize human blood mononuclear cells after OROV in vitro infection, starting with T, B and monocytes cells lineage models. Jurkat cells are T cells lymphoblasts, from acute T cell leukemia, which have been used to elucidate the mechanisms behind the TCR signaling and interleukin production, such as IL-2. The variety of receptors found in Jurkat cells, like TCR-CD3 molecules, CD4, C-X-C chemokine receptor 4 (CXCR4) and C-C chemokine receptor type 5 (CCR5), have also helped the elucidation of many steps of the human immunodeficiency virus (HIV) replication cycle [41]. In our results, we observed that OROV can infect and more efficiently replicate in T CD4^+^ lymphocytes (Jurkat cells), corroborating with previous studies [42], than other lymphoid lineages. Monocytes THP-1, which are a characterized cell line with phagocytic and lysozyme production capacity, were also susceptible to OROV infection but not as much as Jurkat cells. However, we observed that both cells can keep the virus replicating until at least three passages, which means subculturing for 10 to 12 days. Although in this study we did not evaluate monocytes differentiation, Geddes and collaborators (2018) [42] showed that THP-1 cells differentiated into a macrophage-like cell with PMA activation, are more susceptible to OROV infection.

Our attempt to evaluate human peripheral blood cell infection in vitro led to some interesting findings, despite the difficulties to obtain enough cells for the RNA PrimeFlow™ assay and flow cytometry analysis. We observed a few CD3^+^ lymphocytes carrying viral genome throughout 24 hpi and the percentage of positive cells slightly increased over time. We also attempted to investigate monocytes by tracking CD14^+^ cells and the results were similar. Other populations such as DCs and macrophages have been studied in immunodeficient animal models, showing a productive infection of OROV when the IFN signaling pathways are compromised [18]. In our results, we showed evidence of OROV infection in human peripheral blood DCs in vitro; however, we did not further investigate cells’ activation profile. Moreover, we were not able to follow up human B cells infection, but a small percentage of B cells were positive for the OROV genome by flow cytometry, at 24 hpi. In fact, Jeko-1 cells were also susceptible to OROV productive infection but to a lesser extent than Jurkat cells. Additionally, we observed only a few Jeko-1 cells positive for the OROV genome and proteins 48 hpi, while there was no evidence of the antigenome production, suggesting that these cells can hamper viral genome replication, but the genome may be maintained for longer periods in the cells. To verify this hypothesis, we also subcultured Jeko-1 infected cells, which resisted more than three passages.

Whilst OROV NSs protein demonstrated to be an IFN antagonist in a previous study with human lung adenocarcinoma cells (A549) cells [12], these cytokines are still protective in mice models preventing viral replication in the SNC [18,40]. We observed that the expression of type I, II and III IFNs increase especially after 24 and 48 hpi in human PBMC infected in vitro. It was associated with an increase in the expression of Mx1 and IFIT1, and to a decrease in viral titers in the supernatant. In addition, when the blocking type I IFN receptor (IFNAR), viral titers in the supernatant increase.

TLR-independent cytosolic recognition systems are important mechanisms in the recognition of RNA viruses that trigger the activation of transcription factors and I IFN expression [13]. In this work, we observed a slight increase in the expression of RIG-I, also known as DDX58, and a significant fold increase in the expression of MDA5 at 12 hpi (*p* < 0.0001), two important proteins that participate in the recognition of viral RNA in the cytoplasm. Studies performed in mice models have shown that signaling through MAVS and IRF3, IRF7 and IRF5 transcription factors are determinants to contain the progression of OROV infection [18,43]. Our results of PBMCs infection and gene expression analysis by RT-qPCR show evidence that corroborate with mouse model studies and demonstrate that the early response mediated by IFN signaling pathways and ISGs are important to suppress viral replication in human PBMCs.

OROV detection in peripheral blood from patients was recently demonstrated, but there is still a lack of evidence in the literature of the specific susceptible populations [3]. Here we observed that even though OROV was able to infect and replicate in human monocytes, T CD4^+^ lymphocytes and B lymphocytes lineages, its potential to infect and replicate in human PBMC from healthy donors is limited. Taken together, our results show evidence of human PBMC susceptibility to OROV infection in vitro, which are in agreement with the increased expression of IFNs and ISGs observed. Consistently, blocking of IFN response allowed the virus to replicate in vitro, implying that in an immunodeficient environment in vivo OROV could infect and replicate in PBMC populations. Then, in a situation such as a compromised blood–brain barrier as previously reported [7], cells carrying the OROV genome may act as a trojan horse in specific situations.

## 5. Conclusions

The interest to study OROV infection and disease progression increased during the last few years, in parallel with the increasing reports of OROV fever in the Amazonian region and different regions of Brazil. Our study with human blood mononuclear cells brings evidence of OROV capacity to infect these cells and presents new methodologies to study OROV replication by targeting viral genome and antigenome. In addition, we show that human PBMC from healthy donors are susceptible to the infection in vitro; however, OROV capacity to replicate in these cells is limited, and the IFN response and cytokines released are important to block viral replication. Among the studied cells, T CD4^+^ and DCs cells may be an important objective of study in future experiments. Therefore, further studies must be performed to verify the trojan horse possibility, since the OROV genome was found to be kept inside monocytes, B and T cells.

## Figures and Tables

**Figure 1 viruses-12-00785-f001:**
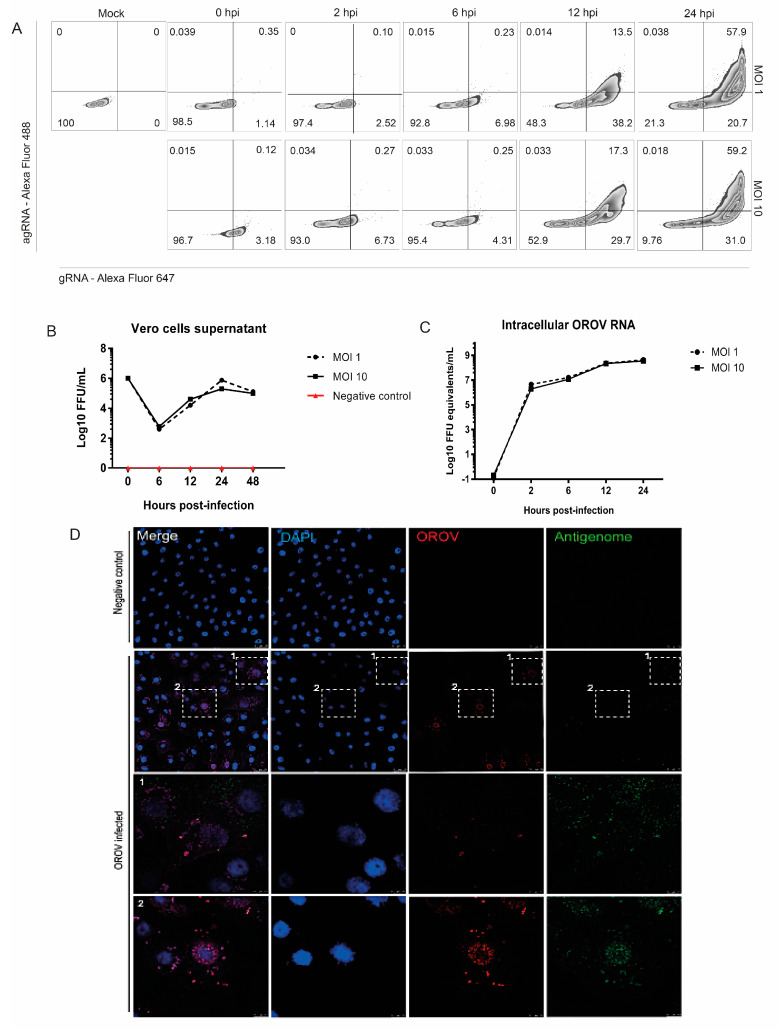
OROV genome and antigenome detection in Vero cells. RNA PrimeFlow™ assay for the detection of OROV productive infection was standardized in Vero cells. (**A**) Vero cells were previously infected with OROV and analyzed by RNA PrimeFlow™ assay at 2, 6, 12 and 24 hpi. Quadrants exhibit OROV genome (gRNA), antigenome (agRNA) or both detection in Vero cells infected at a Multiplicity of Infection (MOI) 1 and 10, and the percentage of events. Noninfected cells were also submitted to the protocol and are shown as the mock sample. (**B**) Vero cells were infected with OROV at MOI 1 and 10. Supernatants were collected at 0, 2, 6, 12, 24, and 48 hpi, and analyzed by focus forming assay (FFA). (**C**) Lysates from cellular monolayers were collected and analyzed by qRT-PCR for the assessment of OROV RNA. (**D**) Vero cells were cultivated in coverslips and infected with MOI 2. At 18 h post-infection, the coverslips were submitted to RNA PrimeFlow™ protocol for immunofluorescence. It is shown OROV proteins in red (Alexa Fluor 594), antigenome in green (AlexaFluor 488) and cell nuclei in blue (DAPI). Images with 63x times magnification. Scales at 2 μm, with 10 μm in zoomed images.

**Figure 2 viruses-12-00785-f002:**
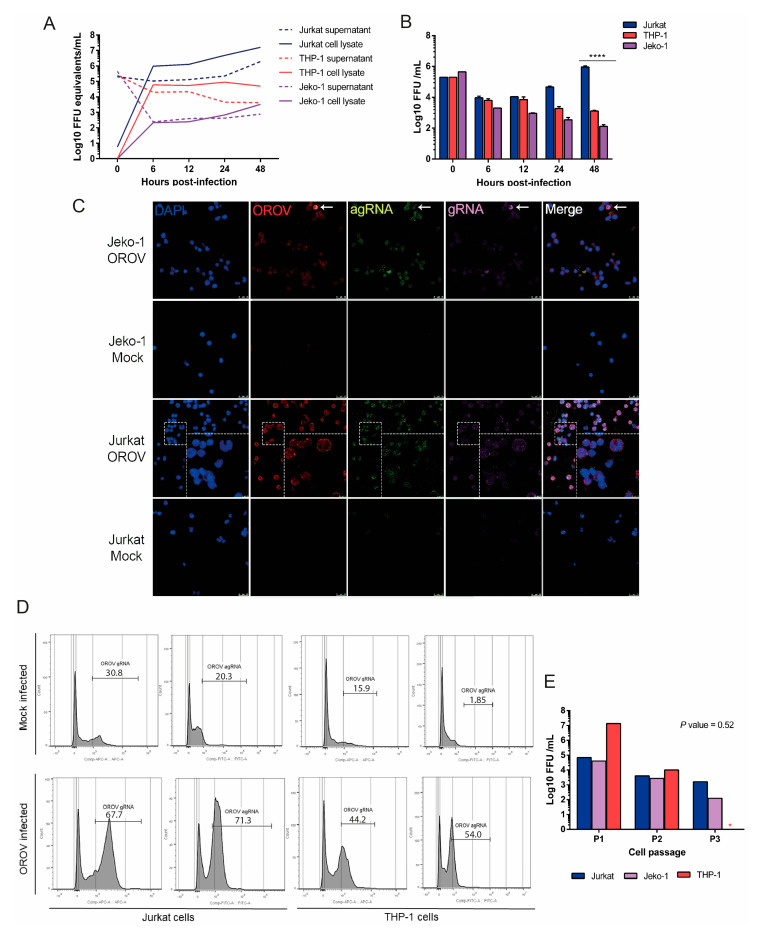
OROV productively infect different human leucocyte lineages. Jurkat, THP-1 and Jeko-1 cells were seeded in 24-well plates and infected with OROV. Cells, supernatants and cell lysates were collected for evaluation by different methodologies. (**A**) Infection at MOI 1 and OROV RNA measured in cellular lysates and supernatants by qRT-PCR for OROV segment S. Lines represent the mean of viral load ± SEM (**B**) FFA analysis of the supernatants. Jurkat cells’ productive viral replication was significantly different from THP-1 and Jeko-1 cells at 48 hpi. Tukey’s multiple comparison test was performed and the *p*-value < 0.0001 is represented by ****. (**C**) Detection of OROV 48 hpi in cells infected at MOI 2 by confocal microscopy. OROV proteins in red (Alexa Fluor 594); genome (gRNA) in magenta (AlexaFluor 647); antigenome (agRNA) in green (AlexaFluor 488); DAPI (blue). Images with 63x times magnification. Scales at 25 μm, with 75 μm in zoomed images. (**D**) Flow cytometry detection of OROV gRNA and agRNA in Jurkat and THP-1 cells infected with MOI 1 at 24 hpi. Mock PF represents the mock infected cells that were submitted to the RNA PrimeFlow™ protocol. (**E**) Jurkat, THP-1 and Jeko-1 cells were infected with MOI 1 and subcultured several times. A sample of the supernatant was collected at each passage (P1, P2, P3) and analyzed by FFA. The THP-1 cell culture did not resist until passage 3 (P3; red asterisk). The Friedman test showed no significant difference in viral replication between the three lineages. The results of infection kinetics are representations of two independent experiments.

**Figure 3 viruses-12-00785-f003:**
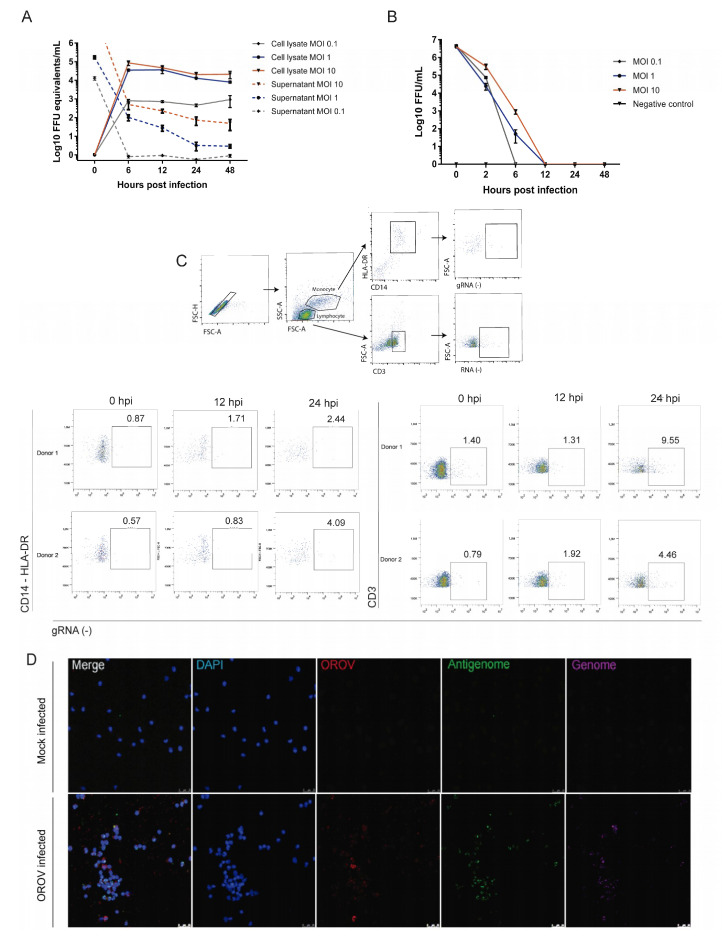
Human peripheral blood monocytes and lymphocytes are susceptible to OROV infection but generate low yields of infectious particles. Human peripheral blood mononuclear cells (PBMCs) were obtained from whole blood of healthy donors, infected in vitro and analyzed by different methodologies. (**A**) PBMCs were seeded in 24-well plates and infected with OROV MOI 0.1; 1 and 10. Cell lysates and supernatants were collected to RNA quantification by qRT-PCR (*n* = 3). (**B**) The supernatants were also analyzed by FFA assay. Symbols represent the mean of viral load ± SEM. (**C**) PBMCs from healthy donors (*n* = 2) were infected with MOI 1, submitted RNA PrimeFlow™ protocol 24 hpi and flow cytometry. CD3^+^ and CD14^+^ HLA DR^+^ percentage of events with OROV Grna, and the gating strategy are shown. (**D**) Detection of OROV 48 hpi in cells infected with MOI 2, by confocal microscopy. OROV proteins in red (Alexa Fluor 594); genome (gRNA) in magenta (AlexaFluor 647); antigenome (agRNA) in green (AlexaFluor 488); DAPI (blue). Images with 63x times magnification. Scales at 25 μm.

**Figure 4 viruses-12-00785-f004:**
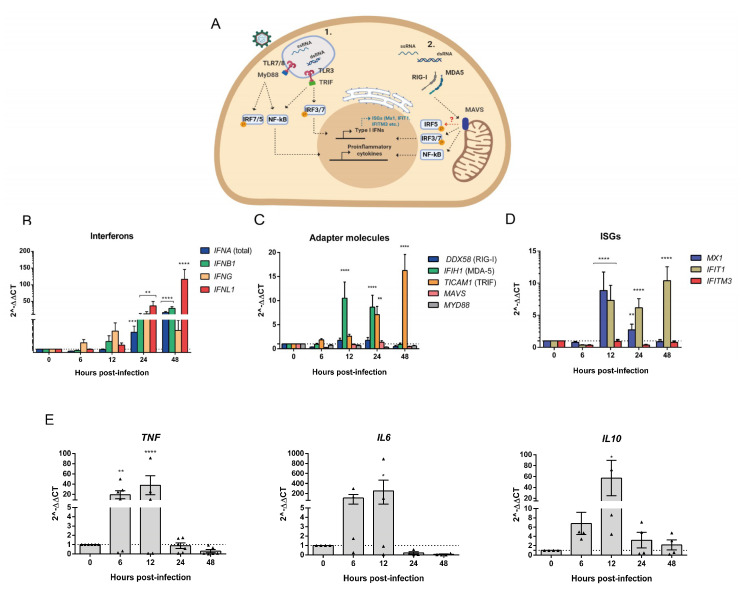
OROV infection of PBMCs induces the expression of cytokines and innate immune response genes. Cell lysates from PBMCs obtained from healthy donors (*n* = 3) and infected with OROV at MOI 10 were processed for RNA extraction and RT-PCR. (**A**) Scheme of the RNA PAMPs recognition pathways analyzed. (**B**–**E**) SYBR Green qRT-PCR data analysis of gene expression are shown in fold change (2^∆∆*C*T). Graphics show results from two independent experiments. Bars represent mean ± SEM. All times post-infection were compared to the time 0 h samples (noninfected) by two-way ANOVA and Dunnett’s multiple comparisons test. **** *p*-value < 0.0001; *** 0.0002 < *p*-value < 0.0007; ** 0.001 < *p*-value < 0.009; and * *p*-value < 0.03.

**Figure 5 viruses-12-00785-f005:**
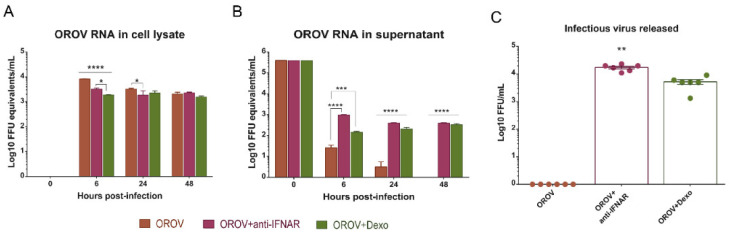
OROV infection and replication in PBMCs is favored when the type I IFN receptor is blocked and when the cells are treated with Dexamethasone. PBMC were pretreated for 2 h with Dexamethasone (Dexo) 1 µM and human anti-IFNAR antibody 5 ng/mL, and infected with OROV at MOI 1. Cell lysates and supernatants were collected for analysis. (**A**) qRT-PCR of cell lysate and supernatant (**B**) for OROV RNA detection through the time of infection. Data were compared by two-way ANOVA and Tukey’s multiple comparison test. (**C**) Infectious particles released in the supernatant were measured by FFA, 48 hpi. Treatment groups were compared by one-way ANOVA and Dunn’s multiple comparisons test. Bars represent viral load ± SEM. **** *p*-value < 0.0001, *** *p*-value < 0.0007; and * *p*-value < 0.05.

**Table 1 viruses-12-00785-t001:** Oligonucleotide sequences for gene expression and viral RNA quantification.

Target	Sequence	Reference
***RIG-1***	F: 5’-TGTGCTCCTACAGGTTGTGGA-3′ R: 5’-CACTGGGATCTGATTCGCAAAA-3′	[22]
***MDA5***	F: 5’-CCAAAGCTGAAGAACACAT-3′ R: 5’-ATCTTCTCTGGTTGCATCT-3′	[23]
***TRIF***	F: 5′-GGCCCATCACTTCCTAGCG-3′ R: 5′-GAGAGATCCTGGCCTCAGTTT-3′	[24]
***MAVS***	F: 5’-GTCACTTCCTGCTGAGA-3′ R: 5’-TGCTCTGAATTCTCTCCT-3′	[25]
***IFNB1***	F: 5’-GCTTGGATTCCTACAAAGAAGCA-3′ R: 5’-ATAGATGGTCAATGCGGCGTC-3’	[26]
***IFNA (total)***	F: 5′-TCCATGAGVTGATBCAGCAGA-3′ R: 5′-ATTTCTGCTCTGACAACCTCCC-3′	[27]
***IFNL1***	F: 5′-CGCCTTGGAAGAGTCACTCA-3′ R: 5′-GAAGCCTCAGGTCCCAATTC-3′	[28]
***IFIT1***	F: 5’-GCGCTGGGTATGCGATCTC-3′ R: 5’-CAGCCTGCCTTAGGGGAAG-3′	[22]
***IFITM3***	F: 5’-ATGTCGTCTGGTCCCTGTTC-3′ R: 5’-GTCATGAGGATGCCCAGAAT-3′	[29]
***MYD88***	F: 5′-ATGGTGGTGGTTGTCTCTGATG-3′ R: 5′-GCATCCTGAGGTTCATCCTGTC-3′	[30]
***GAPDH***	F: 5′- CCCATGTTCGTCATGGGTGT -3’ R: 5′- TGGTCATGAGTCCTTCCACGATA -3’	[31]
***OROV***	F: 5′-TACCCA GATGCGATCACCAA-3′ P: 5′-/FAM/ TGCCTTTGGCTGAGGTAAAGGGCTG/36-TAMSp/-3′ R: 5′-TTGCGTCACCATCATTCCAA-3′	[18]
***IFNG***	F: 5′-TCGGTAACTGACTTGAATGTCCA-3′ R: 5′-TCGCTTCCCTGTTTTAGCTGC-3′	[32]

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
