# Peer review of "Oropouche Virus Infects, Persists and Induces IFN Response in Human Peripheral Blood Mononuclear Cells as Identified by RNA PrimeFlow™ and qRT-PCR Assays"

_viruses, 2020, doi:10.3390/v12070785_

Round 1
Reviewer 1 Report
In the manuscript “Oropouche virus infects, persists and induces IFN response in human peripheral blood mononuclear cells as identified by RNA PrimeFlow™ and qRT-PCR assays” the authors describe in vitro and ex vivo infections of OROV on leukocytic cell lines and PBMCs. These data show the infection of OROV in these cells, however little replication or infectious virus particles are detected. Additionally, the suppression of the interferon pathways can increase the replication of OROV in these cells and the author suggest a potential for higher pathogenesis in immunocompromised individuals. This study is well designed, thorough, and the data agree with the conclusions. This study is of value and interest to the scientific community. Minor suggestions for improvement follow:
- Oropouche virus in title does not need to be italicized because it is not the species name.
- Panel A in figure 1 could be move to supplemental
- Panel A in figure 2 is redundant and does not add any new data.
- Figure 2E. Show histogram information such as percent positive cells.
- Line 359-360 does not refer to the correct figure. Are there stats associated with “consistently higher percentage” of infected Jurkat cells?
- Lines 390-392. Figure 2 legend. There is no panel (G) in the figure or mentioned in the text.
- Panel A in figure 3 does not add any information and can be removed.
- Line 406 should refer to figure 3E not 3F
- Figure 3. A gating scheme to show differentiation of PBMCs into T cells vs monocytes would be beneficial.
- Lines 407-408. Reference to Figure 4 is very limited and not well described in the text. None of the panels to Figure 4 are described or referenced in the text, therefore it is recommended this figure be moved to supplemental information.
- Figure 5. The shades of grey are difficult to distinguish, suggest using patterns. Also, strongly suggest adjusting where the split scale is located. The error bars are not shown or distorted when near the split.
- Lines 541-543. “our results show…” please specifically state which results you are referring to. Increased expression of what?
- Line 548. IgG in CSF is common and not indicative of OROV replication in the central nervous system. Reference to IgG should be removed.
- Lines 555. “compromised blood-brain barrier as previously reported…” please reference the previous study.
- Add p-values to text where appropriate significant comparisons are indicated.
- Add sample size data to each figure where appropriate.
Author Response
"Please see the attachment"

Reviewer 2 Report
Summary
This manuscript investigates the potential of OROV, a clinically important emerging arbovirus, to infect human PBMCs in vitro and characterises the innate immune response to infection. In addition, the study evaluates and applies an innovative technology (RNA PrimeFlow) to address questions in OROV pathogenesis.
The main contributions of the study are:
- A demonstration that RNA PrimeFlow, an innovative technique based on RNA hybridization, adequately differentiates between cells in which OROV is internalized from those in which OROV is able to replicate, partly through the detection of anti-genomic RNA. To the best of this reviewer’s knowledge this has not been published before for OROV. As the authors highlight, anti-OROV antibodies are not easy to obtain therefore RNA PrimeFlow provides an alternative that makes localisation experiments possible.
- Evidence that OROV can infect human blood mononuclear leukocytes (more so in T-cells than B-cells or macrophages) and that viral genome is maintained in cells even after cellular division. T CD4+ and dendritic cells are susceptible and permissive to OROV infection in vitro.
- Evidence that human PBMCs response to OROV infection by upregulation of IFNs and ISGs expression, and that OROV can persist in different types of leukocytes in association with an effective innate immune response.
- OROV replication can be induced in PBMCs by blocking the IFN pathway. The authors suggest that in this way, leukocytes may serve as a trojan horse in specific microenvironments during events of immunosuppression. This hypothesis is very interesting to consider as in rare cases neurological complications develop following OROV infection.
Although OROV has been detected in patient buffy coats previously, no studies have yet been published on which populations of cells are susceptible, the capability of OROV to persist or replicate in human PBMCs, or the PBMC innate immune response. Therefore, this manuscript advances knowledge of OROV pathogenesis and makes insightful suggestions for future research in the area.
Broad comments
The introduction is very good in general, relevant to the subject matter whilst remaining concise. It could be improved by expanding a little the paragraph starting on line 82. For example, there is no mention of the role of NSs protein in antagonising the IFN response, which is known for OROV. This is relevant to the study background therefore I suggest including this briefly in the introduction. E.g. Elaborate on study by Tilston-Lunel et al. 2016, which is already referenced [12]. Additionally, previous work showing that OROV is sensitive to IFNa is also relevant here and would help develop the introduction (https://www.ncbi.nlm.nih.gov/pmc/articles/PMC7114330/)
The scientific content of the results is overall very good and demonstrates a logical and thorough approach in investigating the capability of OROV to infect and persist in PBMCs, the pattern of innate immune response to infection, whilst also evaluating the application of RNA PrimeFlow for investigating OROV pathogenesis. Currently some lack of attention to the labelling of panels within figures between the text and the figures makes it confusing to read and this needs to be fixed. I would like to know if mock-infected controls were included in the Figure 4 experiments, and if so this data should be added to strengthen the results.
I have a concern about Figure 2 Panel D (confocal image) which I have provided detail about in my specific comments. I suspect it may simply be a mis-labelling problem, but if not the presence of virus in the mock samples required explanation.
The discussion adequately summarises the main findings of the study in the context of previous work whilst highlighting the paucity of studies in this area. The paragraph starting line 535 could be improved by discussing previous work showing that OROV is sensitive to IFNa (https://www.ncbi.nlm.nih.gov/pmc/articles/PMC7114330/). The authors suggest a possible ‘trojan horse’ role of PBMCs, which has previously been suggested for OROV and is supported by their data. This would be an interesting area of future study and is likely to be of interest to readers, as the rare instances of severe Oropouche fever often involve virus present in the CNS and the mechanism behind this is not understood.
I am sympathetic to the fact that English may not be the first-language of the authors, however the manuscript does need some work in this area. For example, ‘hybridization’ is spelt inconsistently throughout and needs correcting. I have indicated other specific changes regarding spelling and grammar however authors should carefully check spelling and grammar throughout the manuscript.
Specific comments
Introduction
Minor language/grammar edits:
Line 63: Culicoides paraeneses should be Culicoides paraenesis
Line 70: Change ‘protect’ to ‘protects’
Line 73/74: Change ‘patterns recognition receptors’ to ‘pattern recognition receptors’ and ‘pathogens associated molecular patterns’ to ‘pathogen associated molecular patterns’
Line 84: Change ‘NF-kB e CD80’ to ‘NF-kB and CD80’
Line 100: Change ‘that viral genome is still been found’ to ‘that viral genome was still found’
Materials and methods
Minor language/grammar edits:
Line 122: Change ‘Fetal Bovin Serum’ to ‘Fetal Bovine Serum’
Line 151: Suggest revising ‘Jurkat cells were gently donated’ to ‘Jurkat cells were kindly donated’
Line 246: Change ‘Infection kinetics in Vero cells was also’ to ‘Infection kinetics in Vero cells were also’
Line 248: Change ‘human PBMC infected in vitro was submitted’ to ‘human PBMC infected in vitro were submitted’
Line 261: Change ‘in 24 wells plates’ to ‘in 24 well plates’
Line 282: Write out PMN in full since this is the first use in the manuscript.
Results
Figure 1: Suggest increasing font size of numbers in panel A as almost unreadable in present form. Looks like panel C and D descriptions need to be swapped in the figure legend as they currently don’t correspond to the correct graph.
Figure 2: Please check the panel labelling in the text compared to the figure, I think they have been mixed up, which made this section very difficult to follow. Eg. I think Panel 2D referred to in the text is actually 2F in the figure.
I’m not sure that panel A is necessary, since it is identical to Figure 1 Panel A.
Panel D (confocal image, 2E in text): There is OROV RNA and protein staining present in Jeko-1 Mock and Jurkat Mock fields. Is this correct? If so can the authors explain what the mock control was (I thought it was uninfected Vero cell supernatant, in which case I wouldn’t expect to see this viral staining present). Is the panel labelling in fact incorrect? As I notice THP-1 cells are referred to in the text as being in this panel, which is not the case according to the labelling. Please correct if so as this figure doesn’t currently make sense to me.
Figure 3: Panel D, please increase font size of times (0, 12, 24 hr). Legend states 3 donors, but data only shows 2, please address. Panel E is incorrectly referenced in the text as Figure 3F (line 406), please fix.
Line 371: ‘Moreover, since a high proportion of apoptotic cells were 372 observed after 24 hours of incubation (Supplemental Figure S2)’. I cannot see this data in Fig S2. Please add to figure or amend manuscript text.
Line 404: ‘leukocytes infected cells from PBMCs’ doesn’t make sense, please amend.
Minor language/grammar edits:
Line 341: Change ‘in a immunodeficient mice model’ to ‘in an immunodeficient mouse model’
Line 351: Change ‘intracelular’ to ‘intracellular’ (please correct this throughout manuscript)
Line 367: Change ‘healthy donnors’ to ‘healthy donors’ (also on line 436)
Line 393: Change ‘diference’ to ‘difference’
Line 396: Change ‘suppernantant’ to ‘supernantant’
Line 413: Change ‘respectivly’ to ‘respectively’
Line 414: Change ‘transcripition’ to ‘transcription’
Line 428: Change ‘pro-inflamatory cytokynes’ to ‘pro-inflammatory cytokines’
Line 432: Change ‘infectious’ particles’ to ‘infectious particles’
Line 456: Change ‘controling’ to ‘controlling’
Line 467: Change ‘incluing’ to ‘including’
Line 476: ‘for cDNA obtention’. Do not understand the meaning of the word ‘obtention’.
Line 483: Change ‘when the cells are treated with Dexametazone’ to ‘when the cells are treated with Dexamethasone’
Discussion
Line 495: ‘OROV publications have increased in the last few years, as the number of arboviruses reports have also increased [7,8]’. Please check that reference 7 is the correct one to support this statement.
Line 517: I think ‘evaluate monocytes differentiation, Emmanuel and collaborators (2018) [42]’ should say ‘evaluate monocytes differentiation, Geddes and collaborators (2018) [42]’
Minor language/grammar edits:
Line 495: Change ‘as the number 496 of arboviruses reports have also increased’ to ‘as the number 496 of arbovirus reports have also increased’
Line 502: Change ‘corroborating with previous study’ to ‘corroborating with a previous study’
Supplementary material
Figure S2: Although statistical analysis is referred to in the figure legend, no indication of significance is present on the graphs. Please add.
‘Graphics show the media and standard error (SEM).’ Should this say median?
References
Please check authors carefully, I think there may be a problem stemming from use of bibliographic software. For example, for reference 3, I think ‘Kleber L’ and ‘Luna DS’ are the same author (‘Kleber L’ should be removed). This is also true for reference 42 (Emmanuel V, and Geddes V are the same person).
